# Progressive Feature Interaction Search for Deep Sparse Network

**Chen Gao[1], Yinfeng Li[1], Quanming Yao[1,2], Depeng Jin[1], and Yong Li[1]**
[1]Beijing National Research Center for Information Science and Technology,
Department of Electronic Engineering, Tsinghua University
[2]4Paradigm Inc.
`liyong07@tsinghua.edu.cn`

## Abstract

Deep sparse networks (DSNs), of which the crux is exploring the high-order feature interactions, have become the state-of-the-art on the prediction task with high-sparsity features. However, these models suffer from low computation efficiency, including large model size and slow model inference, which largely limits these models' application value. In this work, we approach this problem with neural architecture search by automatically searching the critical component in DSNs, the feature-interaction layer. We propose a distilled search space to cover the desired architectures with fewer parameters. We then develop a progressive search algorithm for efficient search on the space and well capture the order-priority property in sparse prediction tasks. Experiments on three real-world benchmark datasets show promising results of PROFIT in both accuracy and efficiency. Further studies validate the feasibility of our designed search space and search algorithm.

## 1   Introduction

The deep sparse network (DSN) [9, 4] is a special type of deep networks, which targets learning from sparse and categorical features. Such a learning problem frequently appears in many industrial applications. Take the click-through rate (CTR) prediction as an example, features such as user id, user age, item id, and item category, are usually sparse and high-multidimensional [4]. In advertisement recommendation [27], there are sparse features such as category, brand, etc. In fraud detection [23], there are sparse features such as income level, region, etc.

A general DSN framework (see Figure 1) consists of three layers. First, the embedding layer transforms the raw sparse features into dense embeddings. Then the feature-interaction layer can construct cross-features of raw features' embeddings via feature-interaction operation. The cross-features are further fed into the output layer to calculate the prediction score. To achieve high prediction accuracy, researchers found that the crux of the deep sparse networks is to design complex feature-interaction layers [9, 31, 5, 18, 25].

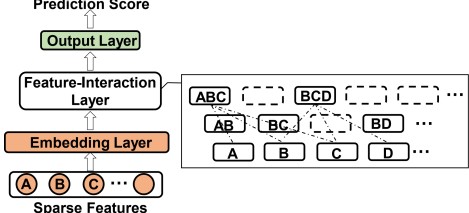

Figure 1: Illustration of deep sparse networks and feature-interaction layer.

For example, in Figure 1, the second-order cross-feature, `AB`, is constructed by two raw features, `A` and `B`, via feature interaction. DeepFM [9] feed the two-order cross-features into multi-layer perceptions for implicitly generating high-order cross-features. Further advances replace the MLP in DeepFM with more complex neural networks, including deep cross network in xDeepFM [18], outer product network in PNN [25], self-attention network in AutoInt [31], and logarithmic transformation

35th Conference on Neural Information Processing Systems (NeurIPS 2021).

network in AFN [5]. Since these deep neural networks work as a black box, high-order cross-features are learned in an implicit manner.

Neural architecture search (NAS) [8, 39], has become a powerful approach in replacing human experts in proposing both effective and lightweight models for various areas such as computer vision [?], natural language processing [29], etc. More recently, there are some works that adopt neural architecture search for designing the feature-interaction layer in DSN. For example, SIF [40] searches simple first order interaction function between users and items; AutoFIS [20] and AutoGroup [19] explicitly search for more powerful higher order cross-features. Although the searched feature-interaction layer is more lightweight to some extent, these methods are suffering from high search costs, resulting in limited prediction performance and application value. However, these works require extremely large search costs, limiting their application values on large datasets.

In general, the human-designed DSNs are faced with the efficiency issue since they mainly pursue prediction accuracy. In fact, efficiency has a very high impact on the application value of machine learning models in real-world scenarios [3, 33, 28]. Considering that today's machine learning systems are always deployed at diverse hardware platforms, inference time and model size are the two most significant factors. Specifically, the inference time can largely affect the time delay of the user services, which largely affects the application value. The model size can determine whether the model can be deployed at specific devices, which sets the constraint to the model's application scenarios, especially for edge devices. Therefore, these complicated deep sparse networks are still limited due to their inferior efficiency.

In this work, to address the above-mentioned efficiency-accuracy dilemma of state-of-the-art deep sparse networks, we propose to utilize NAS to design lightweight and effective models. However, this problem suffers from two challenges. exponentially with the order of cross-features and searching in this space is very hard and time-consuming. Second, in sparse prediction tasks, there exists the order-priority property, which is hard to handle. Specifically, the lower-order cross-features should be assigned higher priority compared with high-order ones. To tackle them, we propose a PROFIT method (short for **PRO**gressive **F**eature **InT**eraction Search). We first design a distilled search space, a low-rank approximation of the full search space, which can vastly reduce the difficulty of finding desired models. More precisely, the full search space, which uses a tensor to represent all possible architectures, is approximated by the composition of low-rank tensors. We then develop a progressive search algorithm that searches cross-features from lower orders to higher orders sequentially. When searching for the higher orders, the corresponding architecture parameters of lower orders are fixed, helping distinguish the various importance of different orders. We conduct experiments on three real-world benchmark datasets and compare both accuracy and efficiency with state-of-the-art models. Empirical results demonstrate that the model searched by our PROFIT can achieve high accuracy while keeping low computation cost and model size. Further studies verify the feasibility of each design, consisting of the search space and search algorithm.

## 2   Related Work

### 2.1   Neural architecture search (NAS)

Neural architecture search (NAS) [42, 1, 39] replaces human experts in designing neural networks architectures, which has been demonstrated a promising approach in many research areas, e.g., vision and natural language processing. There are two significant perspectives that determines its utility: *search space* and *search algorithm*. The *search space* is defined as the set of all possible architectures. It is task-specific and should be general enough to cover existing models but not too general to incur high search costs. The *search algorithm* attempts to find the desired architecture in a given space. A good search algorithm should well explore domain properties to find good architectures and achieve high search efficiency. More recently, some pioneer works [10, 2] propose NAS methods for automatically designing lightweight models for computer vision tasks, which can reduce the inference time on edge devices. Compared with the existing works, our work approach the problem of searching for deep sparse networks, of which there exist critical challenges of large search space and high search cost.

## 2.2 Deep Sparse Networks (DSNs)

As mentioned in Section 1, deep sparse networks (DSNs) serve as the state-of-the-art solution for the sparse prediction task. Its crux is to extract cross-features, which can have strong abilities in prediction, from raw input features.

Here, for the sparse prediction task, we recap the evolution from earlier shallow models to DSNs. Factorization machine [27] is a pioneer work that assigned a linear matrix factorization operation to all possible second-order cross-features (built with two raw features), which was extended by NFM [13] and AFM [35]. To overcome the shortcomings of these shallow feature-interaction operations, deep sparse networks [9, 18, 5, 31, 25] are proposed. In general, deep sparse networks adopt neural feature-interaction layers to extract cross-features, as illustrated in Figure 1. Wide&Deep [4] and DeepFM [9] feed the low-order features into multi-layer perceptrons (MLPs) to implicitly reveal any order cross-features. More complex neural networks are proposed to serve as the feature-interaction layer, replacing MLPs, such as the compressed interaction network (CIN) in xDeepFM [18] and outer product network in PNN [25].

There are some recent works [31, 5] approaching the cross-feature search of DSNs. Specifically, in AutoInt [31], the feature-interaction layer is designed to stack multi-layer self-attention architectures; AFN [5] used a logarithmic transformation layer to convert the order to a learnable parameter and then adopted a feature summation as the feature-interaction layer, which fuzzes the order of cross-features.

However, there is no model that can achieve both promising efficiency and high accuracy. Thus, it is important to solve the efficiency-accuracy dilemma in DSNs. It is worth mentioning that there are some recent works [22, 41] that adopt AutoML in feature embeddings, which is the input of DSNs, for reducing the parameter number of embeddings.

## 2.3 Deep Network Sparsification

Besides, it is worth mentioning there is another research direction closely related to this work, network sparsification [26, 34, 11, 37], which aims to prune the network components or parameters to reduce the network size while keeping a little performance decrease. Different from them, our work searches for the architecture in an interpretable and explicit manner: finding architectures that reserve important cross-features and remove useless ones; for network pruning models, embedding parameters are dropped in an implicit manner — the pruning masks are always implicitly learned from data, does not provide any interpretability. In deep sparse prediction tasks, it is more significant to explicitly lighten the model at the level of cross-feature, while existing works of network sparsification or network pruning is at the level of latent dimension.

## 3 The Proposed Method

### 3.1 The Search Problem

Existing works of DSNs [18, 31, 5] have shown that the feature-interaction layer is an important component. It can be defined as a network module of which the input is embeddings of raw features, and the output is interactions of embeddings. The interactions have extracted features' co-relations and can then be fed to the output layer for obtaining the final prediction results. A feature-interaction layer that extracts $o$-th order cross-features can be formulated as

$$\mathbf{Layer}_o = \big\{ \mathbf{f}_{c_1} \odot \mathbf{f}_{c_2} \cdots \odot \mathbf{f}_{c_o} | c_o \in \mathcal{C}_o \big\}, \text{ s.t. } |\mathcal{C}_o| \leq N_o, \tag{1}$$

of which $\mathbf{c}$ denotes a $l$-size vector of involved features' indexes of a $o$-th cross-feature ($c_o, c_2, ..., c_o$ are indexes of raw features), $\mathbf{f}$ denotes the embeddings of raw features (*i.e.*, the output of embedding layer), and $\mathcal{C}_o$ denotes the set of $o$-th order cross-features on which there is a sparsity constraint of $N_o$, ensuring both the quality of cross-features and the model efficiency. The feature interaction is perturbation invariant, i.e., $\mathbf{f}_1 \odot \mathbf{f}_2 = \mathbf{f}_2 \odot \mathbf{f}_1$. The element-wise product is the most used operation for feature interaction [18, 31, 5]. Let $\mathbf{F} = \{\mathbf{f}_1, \mathbf{f}_2, \cdots, \mathbf{f}_M\}$. Then, in short, the general objective of DSN can be formulated as follows,

$$\min_{\mathbf{F}} \mathcal{L}\big(\phi(\mathbf{F}, \{\mathcal{C}_o\}); \mathcal{D}\big), \tag{2}$$

where $\mathcal{L}$ denotes the loss function and $\phi$ is the prediction function, of which a linear predictor or MLP can be used. The objective of the AutoDSN problem should be finding good sets of feature-interaction

cells, $\{\mathcal{C}_o\}$, with $o$ from 1 to $O$, formulated as follows.

$$\{\mathcal{C}_o\}^* = \min_{\{\mathcal{C}_o\}} \mathcal{L}\big(\phi(\mathbf{F}^*, \{\mathcal{C}_o\}); \mathcal{D}_{\text{val}}\big), \text{ s.t. } \begin{cases} \mathbf{F}^* = \arg\min_{\mathbf{F}} \mathcal{L}\big(\phi(\mathbf{F}, \{\mathcal{C}_o\}); \mathcal{D}_{\text{tra}}\big), \\ |\mathcal{C}_o| \leq N_o, \quad \forall o = 1, 2, ..., O, \end{cases} \tag{3}$$

where $\mathcal{D}_{\text{tra}}$ and $\mathcal{D}_{\text{val}}$ denote the training and validation datasets, respectively.

For the AutoDSN problem, it is a challenging problem due to multiple aspects. First, the search space for $\mathcal{C}$ is relatively large, which burdens the search cost. Second, the desired sets are discrete, and there are also sparse constraints, making it an NP-hard problem. Third, the desired cross-features are closely co-related with each other, from lower orders to higher orders, which should be carefully considered.

### 3.1.1 Revisiting Existing Approaches

For the search problem in (3), an intuitive NAS-based solution is to use a parameterized vector to represent all cross-features' importance. Such solution was recently proposed by AutoFIS [20], assigning different weights to cross-features and using the continuous relaxation, similar as DARTS [21]. For $o$-order cross-feature, a $o$-th order tensor $\mathcal{A}$ can be introduced, where $\mathcal{A}_{c_1, c_2, \cdots, c_o}$ represents the weight of a cross-feature involved with raw features $\{\mathbf{f}_{c_1}, \mathbf{f}_{c_2}, \cdots, \mathbf{f}_{c_o}\}$. Then, a feature-interaction layer in (1) is expressed as

$$\sum_{c \in \mathcal{C}_{\text{full}}} \mathcal{A}_{c_1 c_2 \cdots c_o} (f_{c_1} \odot f_{c_2} \odot \cdots \odot f_{c_o}), \tag{4}$$

where $\mathcal{C}_{\text{full}}$ denotes the set of all possible cross-features. Then the NAS problem with a continuous space can be re-formulated as follows,

$$\mathcal{A}^* = \min_{\mathcal{A}} \mathcal{L}\big(\psi(\mathbf{F}^*, \mathcal{A}); \mathcal{D}_{\text{val}}\big), \text{ s.t. } \mathbf{F}^* = \arg\min_{\mathbf{F}} \mathcal{L}\big(\psi(\mathbf{F}, \mathcal{A}); \mathcal{D}_{\text{tra}}\big), \tag{5}$$

where $\psi$ is the prediction function, which first reserves $N_o$ cross-features with the top-$N_o$ weights to meet the sparsity constraint and then feed reserved features to output layer. That is, it is similar with $\phi$ in Definition 1. Such a continuous space can overcome the issue of discrete search space and the learned architecture parameters provide explicit meaning, *i.e.* the importance of cross-features.

It is worth mentioning that AutoInt [31] learned the importance of cross-features implicitly by self-attention layers, which thus is stilled limited. Therefore we try to *explicitly* search for cross-features at the feature-interaction layer. Faced with the challenge of large search space, we approach the problem with the idea of encoding the space into low-rank vectors for reducing the cost.

### 3.2 Proposed Search Method

First let us recap some recent works [20, 23, 19] adopting NAS to help explicitly search powerful cross-features as the feature-interaction layer. Although the searched feature-interaction layer is more lightweight, these methods suffer from high search costs. Furthermore, it ignores the different orders of cross-features, from 1 to $O$, and consider them as the same, which results in a very large-size weight tensor $\mathcal{A}$, with length $\sum_{o=1}^{O} C_M^o$, where $M$ denotes the number of raw features. Such a NAS problem with a continuous space still has two critical issues.

- First, representation of the search space takes too many architecture parameters, which slows down the optimization procedure. With $M$ fields of raw features and the desired feature order $O$, the size of architecture parameter for the search space of cross-feature is close to $C_M^O$, which approximately equals to $M^O$ considering $M \gg O$. Obviously, $M^O$ is a too large number in the gradient-based NAS. For example, the widely-used Criteo benchmark dataset has $M = 39$ of fields of raw features. If we choose the highest order as 4, then $C_{39}^4 = 82{,}251$. Such a parameter number is much larger compared with existing works of gradient-based NAS [21, 36], of which the number would always be fewer than one hundred. Therefore, the challenge of large search space is still unresolved.
- Second, such design still ignores the *order-priority property* in deep sparse prediction tasks. Specifically, the property consists of two characteristics revealed by human practice: higher-order features' quality can be relevant to their de-generated low-order ones, and lower-order cross-features are likely to be more vital compared with higher-order ones.

To address the above-mentioned two limitations in Section 3.1, we present our solution PROFIT here with 1) a *distilled low-rank approximated search space* to address the space issue and 2) a *progressive search algorithm* to take order-priority property into consideration.

### 3.2.1 Distilling the Search Space

As above, the architecture parameter, i.e., tensor $\mathcal{A}$, can be too large and ignore priorities on feature interactions. This can lead to high search costs and not being capable of finding desired architectures. Here, we study the property of $\mathcal{A}$ and attempt to approximate it with much fewer parameters.

We conducted an experiment on the Criteo dataset and set the max order as two due to the computation cost. This dataset has 39 fields, and thus weights $\mathcal{A}$ for all second-order interactions can be represented by a matrix of size $39 \times 39$. Singular values from $\mathcal{A}$ and a randomly generated matrix are compared in Figure 2 (a). We can observe that the parameter matrix has several dominant singular values, exhibits a low-rank property. In other words, skew singular values in Figure 2 (a) shows there is redundancy in the architecture matrix for the second-order cross-feature. Besides, note that feature interaction is perturbation invariant, i.e., $f_1 \odot f_2 = f_2 \odot f_1$, which means the tensor should be symmetric. Thus, we propose to distilled tensor $\mathcal{A}$ as

$$\mathcal{A} \approx g(\{\boldsymbol{\beta}^o\}) \equiv \sum_{r=1}^{R} \underbrace{\boldsymbol{\beta}^r \circ \cdots \circ \boldsymbol{\beta}^r}_{\boldsymbol{\beta}^r \text{ repeats } O \text{ times.}}, \tag{6}$$

where each $\boldsymbol{\beta}^r \in \mathbb{R}^{1 \times M}$, and $R \ll M$ is the positive integer. Equation (6) can also be seen as a symmetric CP decomposition [17] for the tensor $\mathcal{A}$. In other words, $\boldsymbol{\beta}^r$ and $\boldsymbol{\beta}^o$ refer to the approximated tensor and decomposed tensor, respectively. Subsequently, the search problem in (3) is transformed as

$$\{\boldsymbol{\beta}^o\}^* = \min_{\beta} \mathcal{L}\big(\psi(\mathbf{F}^*, g(\{\boldsymbol{\beta}^o\})); \mathcal{D}_{\text{val}}\big), \quad \text{s.t. } \mathbf{F}^* = \arg\min_{\mathbf{F}} \mathcal{L}\big(\psi(\mathbf{F}, g(\{\boldsymbol{\beta}^o\})); \mathcal{D}_{\text{tra}}\big). \tag{7}$$

Comparing with the original $\mathcal{A}$ in (5), of which size is about $M^O$ and grows exponentially with $O$, $\{\boldsymbol{\beta}^r\}$ is of size $M \times O$ and can be much smaller. Thus, the number of architecture parameters is reduced significantly.

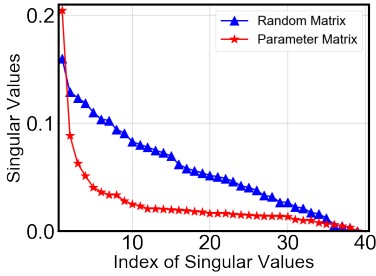

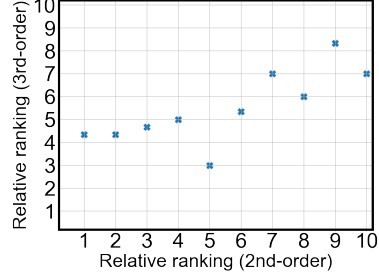

(a) Singular values of $\mathcal{A}$ for the second-order cross-features.

(b) Correlations between low-order and high-order cross-features.

Figure 2: Key observations on search space (left) and search algorithm (right) for PROFIT.

**Remark 1** *Here we discuss a research topic related to our solution, search space compression [6, 38, 12]. Different from these works such as [6] which is based on compressed sensing, our method is based on low-rank approximation. Therefore, although both these works and our work conduct the space distillation, the fundamental assumptions (vector space v.s. matrix space) are different.*

### 3.2.2 Progressive Search Algorithm

As mentioned above, different orders of cross-features are highly correlated. To confirm that higher-order cross-features that are constructed by important lower-order cross-features tend to be important, since the ML1M dataset has fewer fields of features and then a full space of high-order features is easier to optimize, we conduct the experiment on the ML1M dataset. We still use $\mathcal{A}$ in (4) and set the maximum order to two and three. The relative ranking is computed according to the importance weights of all same-order cross-features. For example, a second-order cross-feature with the relative ranking as one means it is the most important second-order cross-feature. For a given 2nd-order cross-feature, $\mathbf{f}_{c_1} \odot \mathbf{f}_{c_2}$, we can also compute the average relative ranking of third-order cross-features that consists of $c_1$ and $c_2$. We study whether an important second-order cross-feature can generate

important third-order cross-features. The relative ranking is shown at Figure 2 (b), and experimental details are in Appendix A.5. We can easily observe that a good 2nd-order cross-feature is more likely to generate a good 3rd-order cross-feature. Note that these points are not monotonically increasing, with some *special* points such as the 5-th one. It shows the necessity of utilizing auxiliary parameters to model the high-order cross-features.

The above motivates us to adopt a progressive strategy to search for the architecture parameters from lower orders to higher orders, to tackle the problem of modeling the order-priority property. The complete steps are in Algorithm 1. Specifically, when searching for $o$-th order cross-feature, at step 2, we propose to fix the architecture parameters with order lower than $o$, i.e., $\{\boldsymbol{\beta}^1, \ldots, \boldsymbol{\beta}^{o-1}\}$. Then, following existing works [21, 20, 40], at step 4 and 5, we also use gradient descent to update both architecture and embedding parameters.

---

**Algorithm 1** Progressive gradient descent.

---

1: **for** $o = 1, \ldots, O$ **do**
2:      fix $\boldsymbol{\beta}^1, \boldsymbol{\beta}^2, ..., \boldsymbol{\beta}^{o-1}$;
3:      **for** $epoch = 1, \ldots, E_o$ **do**
4:          Update architecture parameters $\boldsymbol{\beta}^o$ using gradients $\nabla_{\boldsymbol{\beta}^o} \mathcal{L}\big(\psi(\mathbf{F}, g(\{\boldsymbol{\beta}^o\})); \mathcal{D}_{\text{val}}\big)$;
5:          Update embedding parameters $\mathbf{F}$ using gradients $\nabla_{\mathbf{F}} \mathcal{L}\big(\psi(\mathbf{F}, g(\{\boldsymbol{\beta}^o\})); \mathcal{D}_{\text{tra}}\big)$;
6:      **end for**
7: **end for**

---

### 3.2.3 Retraining the Searched Lightweight Model

Finally, as the existing NAS works [21, 42] reveal, it is better to re-train the model with the obtained embedding parameters. In our task, to ensure both the prediction accuracy and model efficiency, we only reserve the cross-features with top-$N_o$ importance weights to meet the sparsity constraints. The reason is in two folds. First, some features do not benefit the task, and removing them can benefit the training procedure to achieve better prediction accuracy. Second, more importantly, only reserving $N_o$ cross-features can largely reduce the model size and inference times, ensuring the real-world deployment on various hardware platforms.

## 4 Experiments

### 4.1 Settings

**Datasets.** To validate the effectiveness of our proposed PROFIT, we conduct experiments on three benchmark datasets (Criteo, Avazu and ML1M) widely used in existing works of deep sparse networks [5, 31, 18] to evaluate the performance, of which the details are provided in the Appendix.

**Baselines.** We compare our PROFIT with state-of-the-art methods for sparse prediction. We choose methods that are widely used and open-source:

- First-order method: logistic regression [14] (denoted as "LR");
- Second-order methods: we compared with the most widely-used second-order method factorization machine [27] (denoted as "FM") and the state-of-the-art second-order method attentive factorization machine [35][1] (denoted as "AFM"). NFM [13] is a de-generated version of AFM, and thus we do not compare with it.
- Higher-order methods with deep feature-interaction layers: deep factorization machine DeepFM [9] extreme deep factorization machine [18][2] (denoted as "DeepFM") , extreme deep factorization machine [18][3] (denoted as "xDeepFM"), product neural network [25][4] (denoted as "PNN"), automatic feature Interaction learning [31] (denoted as "AutoInt") [5] and adaptive factorization network [5][6] (denoted as "AFN").

---

[1] https://github.com/hexiangnan/attentional_factorization_machine
[2] https://github.com/ChenglongChen/tensorflow-DeepFM
[3] https://github.com/Leavingseason/xDeepFM
[4] https://github.com/shenweichen/DeepCTR
[5] https://github.com/DeepGraphLearning/RecommenderSystems.
[6] https://github.com/WeiyuCheng/AFN-AAAI-20

Table 1: Testing performance on Loss and AUC over three datasets.

| Dataset | | Criteo | | Avazu | | ML1M | |
|---|---|---|---|---|---|---|---|
| Order | Model | Loss | AUC | Loss | AUC | Loss | AUC |
| First | LR | 0.4568 | 0.7936 | 0.3934 | 0.7570 | 0.5880 | 0.7418 |
| Second | FM | 0.4552 | 0.7954 | 0.3809 | 0.7803 | 0.5387 | 0.7938 |
| | AFM | 0.4538 | 0.7969 | 0.3832 | 0.7764 | 0.5530 | 0.7809 |
| High | PNN | 0.4508 | 0.8029 | 0.3798 | 0.7839 | 0.5254 | 0.8042 |
| | AutoInt | 0.4443 | 0.8074 | 0.3786 | 0.7849 | 0.5285 | 0.8014 |
| | AFN | 0.4428 | 0.8094 | 0.3803 | 0.7818 | 0.5294 | 0.8038 |
| | DeepFM | 0.4469 | 0.8050 | 0.3787 | 0.7844 | 0.5318 | 0.8044 |
| | xDeepFM | 0.4430 | 0.8093 | 0.3781 | 0.7856 | **0.5280** | 0.8069 |
| NAS | AutoFIS | 0.4537 | 0.7971 | 0.3815 | 0.7794 | 0.5353 | 0.7956 |
| | PROFIT | **0.4427** | **0.8095** | **0.3735** | **0.7883** | 0.5281 | **0.8077** |

These human-designed baselines models are also selected as the representative DSN models for comparison in the AutoFIS paper. Besides, there is another recent work, i.e., AutoGroup [19], which is similar to AutoFIS. Considering it has close prediction performance and search cost compared with AutoFIS, we only compare AutoFIS in our experiments. There are some recent works [30, 16] approaching the problem via automatically fusing human-designed models, which is still restrained by human-designed feature-interaction layers.

As mentioned above, there are some pioneer works that propose NAS methods for lightweight CNN models for computer vision tasks. Here we do not compare with them since it is hard to adapt them to the deep sparse networks.

**Evaluation Metrics.** We evalua te models from two perspectives, prediction performance and inference efficiency. For accuracy, we use two standard metrics, Logloss (denoted as "Loss") [24] and AUC [7], following [5, 9, 18, 27, 25]. For efficiency, we also use two standard metrics, inference time and model size, which are also widely used in existing works of deep sparse networks [31, 15].

**Hyper-parameter Settings.** To ensure the reproducibility of experimental results, here we introduce the implementation setting, of which more details are in Appendix. We implement our methods using PyTorch. We apply Adam with a learning rate of 0.001 and a mini-batch size of 4096, a widely-used setting in existing works [5, 31]. We set the embedding sizes to 16 in all the models. We use the same neural network structure ({400, 400, 400}) for all methods that adopt MLP for a fair comparison, following [5, 31]. All the other hyper-parameters are tuned on the validation set. To make the results more reliable, for each instance, we run the repetitive experiments three times with different random seeds and report the average value.

## 4.2 Benchmark Comparison

**Prediction accuracy.** In Table 1, we present the prediction performance of all these methods, from which we have the following observations. First, models with higher-order cross-features can achieve better prediction performance than lower-order ones, which verifies the rationality of exploring high-order feature interactions. Second, with the carefully designed search space and search algorithm, PROFIT can achieve the best prediction performance on three benchmark datasets (or very close to the best baseline). Third, the deep sparse networks achieve better performance compared with shallow models, which demonstrates the necessity of designing powerful deep sparse networks. Note that due to the limitations in search space and algorithm, which make the searching procedure very slow, AutoFIS does not achieve promising results compared with human-designed ones[7].

**Efficiency.** We further compare the efficiency and present the results in Figure 3. We can observe that PROFIT is lightweight and fast-to-infer among all compared methods. In real-world applications, both the accuracy and the inference efficiency should be considered. Since our method not only achieves promising accuracy but also reduces the inference time, it has high application value in practice.

---

[7]AutoFIS achieves better performance compared with human-crafted baselines when setting the max order of cross-features as three, as shown by the original paper. However, during the experiments in our normal hardware platform, for the Criteo dataset and Avazu dataset, it cannot be well optimized due to too large search space and computation costs. Thus, we only report the results setting max order to two for these two datasets.

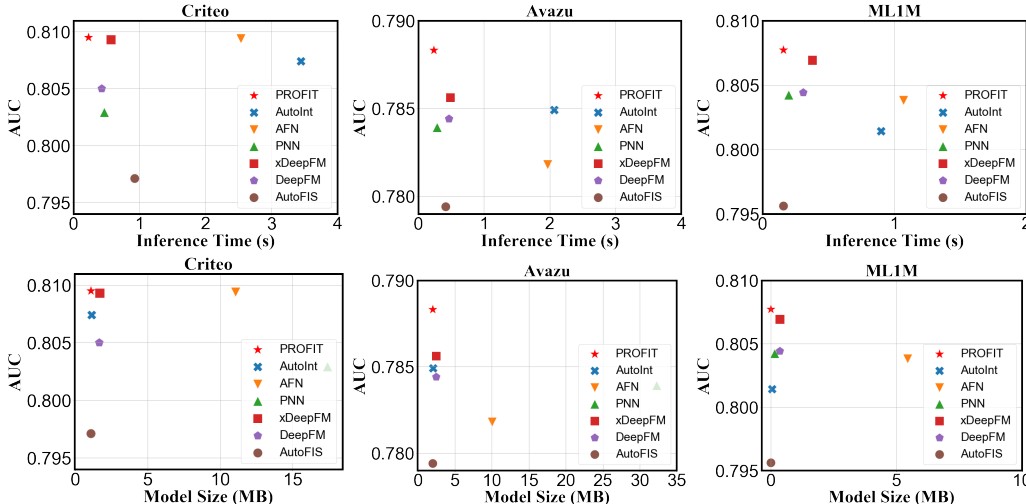

Figure 3: Performance comparison of PROFIT and baseline models. ("Size" refers to the model size without feature embedding layer. "Time" refers to the inference time for one million samples.)

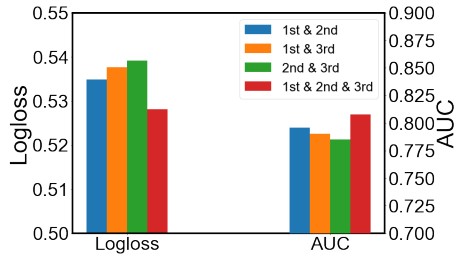

Figure 4: Performance of utilizing different orders' cross-features on ML1M dataset.

Table 2: Performance of searched model with progressive/one-step/random search. Due to lack of space, only AUC is reported.

| Search Algo. | Criteo | Avazu | ML1M |
|---|---|---|---|
| Random with PS | 0.7718 | 0.7499 | 0.7692 |
| One-step Search | 0.8077 | 0.7851 | 0.8053 |
| PROFIT | 0.8095 | 0.7883 | 0.8077 |

## 4.3 Understanding the Search Strategy

**Progressive search.** In order to validate the effectiveness of the progressive manner in our search algorithm, we compared with the performance that utilizes a one-step manner. It can be regarded as the de-generated version of our PROFIT that replaces our progressive search with one step, and the details of its searching procedure are shown in Appendix A.4. We also compare the random search with PS (parameter sharing), of which architecture parameters are randomly assigned, and the model parameters are shared during the search procedures. The final performance of the searched architectures is shown in Table 2. The curve of validation performance during the searching procedure is shown in Figure 5.

From these results, we can observe that random search achieves the worst performance since it does not explore any domain properties, as we have analyzed in Section 3.1. Also, with the simple one-step search algorithm, the performance is poorer compared with the progressive search algorithm. This validates the rationality of our design of progressive search, which can well capture the order-priority property of cross-features in the deep sparse prediction task. In short, the effectiveness of our design of progressive search is well validated by the results.

**Distilled search space.** First, we conduct experiments to study the searched results if different search space is used. We compared our distilled space, i.e., Eqn (7), with a full space, i.e. Eqn (5). We choose the order of three. The curve of validation performance during the search procedure of two large-scale datasets, Avazu and Criteo, is shown in Figure 6, and the final results are shown in Table 3. We can observe that searching in the distilled space can easily reach a promising validation performance. However, when searching in the full space, it requires a higher computation cost, and as a result, it is hard to obtain good architecture parameters. As a result, the distilled search space

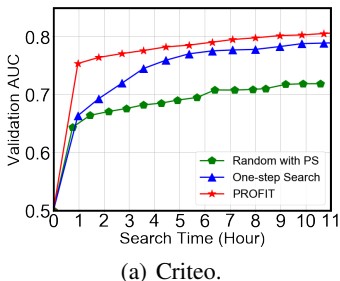 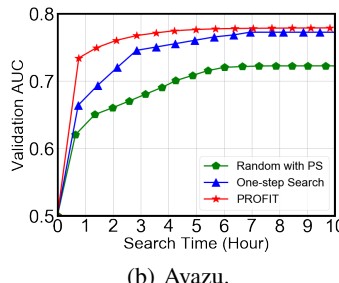

(a) Criteo.    (b) Avazu.

Figure 5: The validation performance during searching with progressive/one-step/random search.

obtains better prediction performance. These results show that with a reduced budget for searching, using the distilled space is a much better choice compared with the original full search space.

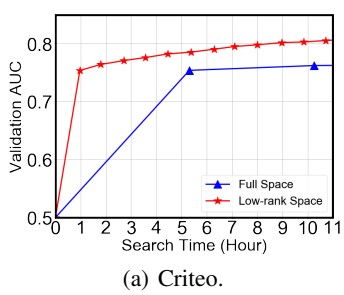 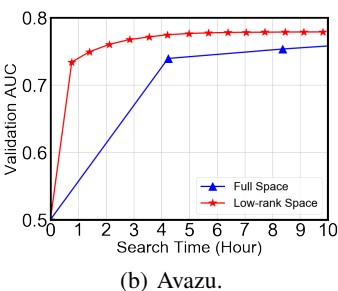

(a) Criteo.    (b) Avazu.

Figure 6: The validation performance during searching with different search space (please note that we use validation performance to illustrate the searching procedure following existing NAS works).

Table 3: Searched model's performance with different search space.

| Dataset | Criteo | | Avazu | | ML1M | |
|---|---|---|---|---|---|---|
| Model | Loss | AUC | Loss | AUC | Loss | AUC |
| Full Space | 0.4533 | 0.7975 | 0.3802 | 0.7807 | 0.5353 | 0.7957 |
| PROFIT's Space | 0.4427 | 0.8095 | 0.3735 | 0.7883 | 0.5281 | 0.8077 |

## 5  Conclusion

In this work, we approached the problem of deep sparse prediction with neural architecture search. We propose a distilled search space which is a low-rank approximation of the full space, significantly reducing the search cost. We then propose a progressive differentiable search algorithm that can efficiently search for desired models in the space while being consistent with the order-priority property. Extensive experiments on three real-world benchmark datasets demonstrate that our proposed PROFIT can achieve both high accuracy and high efficiency. Further studies illustrate that the proposed search space and search algorithm of PROFIT is feasible. As for future work, we will explore more on how dataset characteristics affect the searching results.

## Acknowledgment

This work was supported by National Natural Science Foundation of China under U1936217, 61971267, 61972223, 61941117, 61861136003.

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
