# A APPENDIX

## A.1 Data Preparation

First, we remove the infrequent features (appearing in less than threshold instances) and treat them as "<unknown>" features, where the threshold is set to 10 and 5 for Criteo and Avazu, respectively. Second, to alleviate the impact of large variance in numerical features, following [31], we normalize numerical values by transforming a value $z$ to $log^2(z)$ if $z > 2$, which is proposed by the winner of Criteo Competition[8]. Third, 80% of randomly shuffled data is allotted to training, and we randomly split the rest into validation and test sets with equal size.

## A.2 Datasets

- **Criteo**[9] dataset consists of 45 million clicking records in online advertising systems. The labels in Criteo are whether the user has clicked the item or not.
- **Avazu**[10] dataset is another widely used sparse-feature dataset consisting of about 40 million clicking records on a mobile advertising service.
- **ML1M**[11] is a famous movie-rating dataset consisting of 1 million rating records along with user profiles and movie attributes. The label in ML1M is assigned according to the rating score (the label is assigned to 1 if the rating score is larger than 3 and 0 otherwise).

Table 4: Statistics of the used datasets.

| Dataset | #Samples | #Fields | #Features |
|---------|----------|---------|-----------|
| Criteo | 45,840,617 | 39 | 998,960 |
| Avazu | 40,428,967 | 23 | 1,544,488 |
| ML1M | 1,000,210 | 5 | 6,041 |

## A.3 Implementation Details

All experiments are implemented by PyTorch with codes released at `https://github.com/tsinghua-fib-lab/AutoDSN`, conducted in the following environment:

- Operating system: Ubuntu 18.04.5 LTS
- CPU: Intel(R) Xeon(R) CPU E5-2650 v4 @ 2.20GHz
- GPU: GeForce RTX 2080 Ti
- Software version: Python 3.7.7; Pytorch 1.5.0; Numpy 1.18.3; pandas 1.0.3; scikit-learn 0.22.2

## A.4 One-step Search

The one-step search (Algorithm 2) can be regarded as the de-generated version of our PROFIT that replaces the progressive search with plain gradient descent.

---

**Algorithm 2** One-step search.

---

1: **for** $epoch = 1, \dots, E$ **do**
2:     Update architecture parameters $\{\boldsymbol{\beta}^1, \dots, \boldsymbol{\beta}^O\}$ using gradients $\nabla_{\boldsymbol{\beta}^o} \mathcal{L}\big(\psi(\mathbf{F}, g(\{\boldsymbol{\beta}^o\})); \mathcal{D}_{\text{val}}\big)$;
3:     Update embedding parameters $\mathbf{F}$ using gradients $\nabla_{\mathbf{F}} \mathcal{L}\big(\psi(\mathbf{F}, g(\{\boldsymbol{\beta}^o\})); \mathcal{D}_{\text{tra}}\big)$;
4: **end for**

---

## A.5 Details of Figure 2

For a given 2nd-order cross-feature, $\mathbf{f}_{c_1} \odot \mathbf{f}_{c_2}$, we first compute its relative ranking, denotes as $r_1$ among all 2nd-order cross-features. A smaller relative ranking means a more important cross-feature.

---

[8] `https://www.csie.ntu.edu.tw/~r01922136/kaggle-2014-criteo.pdf`
[9] `https://www.kaggle.com/c/criteo-display-ad-challenge`
[10] `https://www.kaggle.com/c/avazu-ctr-prediction`
[11] `https://grouplens.org/datasets/movielens`

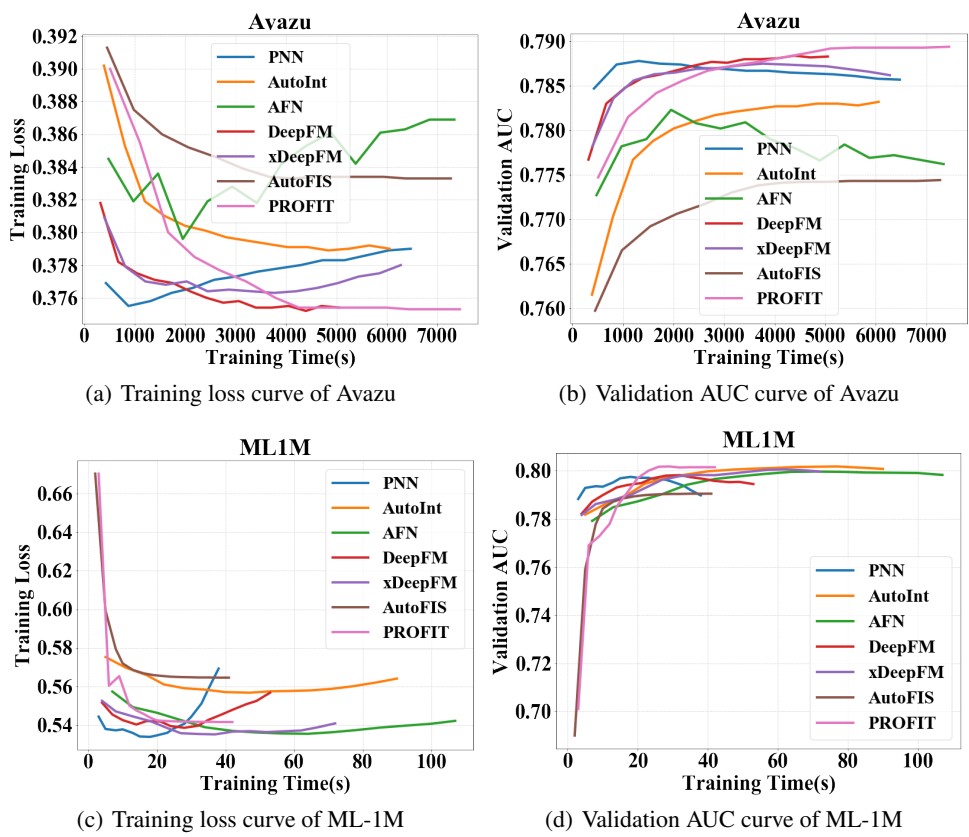

(a) Training loss curve of Avazu

(b) Validation AUC curve of Avazu

(c) Training loss curve of ML-1M

(d) Validation AUC curve of ML-1M

Figure 7: Training curve of baselines and the searched architecture of our PROFIT.

Then we compute the average relative ranking of 3rd-order cross-features that consists of $c_1$ and $c_2$ among all 3rd-order ones, denotes as $r_2$. Note that this relative ranking is completely different from the "rank" in "low-rank".

## A.6   Learning Curves

We also present the training curve of the searched model and human-designed models in Figure 7. We choose two datasets, Avazu and ML1M, due to the limit of space. It can be observed that the searched model by our PROFIT has a similar or smaller training cost compared with other methods, with achieving better validation performance during the training procedure.

## A.7   Ablation Study

**Different ranks.** We also study the prediction performance if we use different ranks in low-rank approximation, and we present the comparison in Figure 8. We can observe that setting rank from 1 to 5 achieves similar performance. It also verifies the rationality of low-rank approximation. Here we show the results of two large-scale datasets due to the limit of space, and ML-1M has similar results and conclusions in our experiments.

**Output Layer.** For the three utilized datasets, we compared PROFIT's performance of utilizing three kinds of output layers, linear predictor, which adopts a weighted sum, and MLP predictor, which feeds the feature interactions to a simple shallow multi-layer perceptron, and hybrid, which fuses the above two at the same time. We present the results in Table 5, and from them, we can observe that different choices of the output layer have a slight impact on the searched model's performance. Linear predictor is good enough for our PROFIT.

In short, the effect of the output layer is far slighter compared with the feature-interaction layer, which validates our motivation.

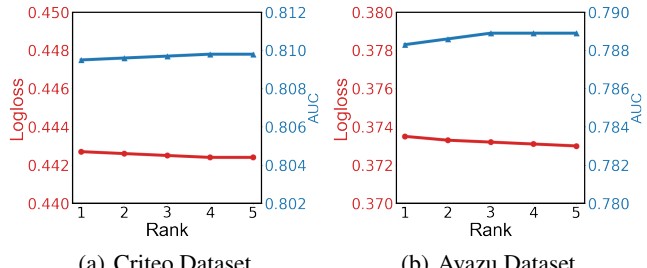

(a) Criteo Dataset        (b) Avazu Dataset

Figure 8: Impact of rank in low-rank approximated search space.

Table 5: Impact of different output layer.

| Dataset | ML1M | | Criteo | | Avazu | |
|---|---|---|---|---|---|---|
| Output Layer | Loss | AUC | Loss | AUC | Loss | AUC |
| Linear | 0.5281 | 0.8077 | 0.4427 | 0.8095 | 0.3735 | 0.7883 |
| MLP | 0.5287 | 0.8075 | 0.4420 | 0.8094 | 0.3736 | 0.7881 |
| Hybrid | 0.5285 | 0.8078 | 0.4427 | 0.8092 | 0.3731 | 0.7885 |

**Embedding Size.** The embedding size may affect the model capacity of the deep sparse network's embedding layer, and here, we study its impact on prediction accuracy. In our above-mentioned experiments, we set the embedding size as 16. We present the results of our PROFIT with different embedding sizes on three datasets in Table 6. The results demonstrate setting the embedding size to 16 is a good choice, although not the best one, for our PROFIT. The impact of embedding sizes is smaller than the different choices of feature-interaction layers.

Table 6: Impact of different embedding sizes.

| | ML1M | | Criteo | | Avazu | |
|---|---|---|---|---|---|---|
| Size | Loss | AUC | Loss | AUC | Loss | AUC |
| 8 | 0.5296 | 0.8080 | 0.4410 | 0.8091 | 0.3715 | 0.7879 |
| 16 | 0.5295 | 0.8088 | 0.4407 | 0.8095 | 0.3721 | 0.7883 |
| 32 | 0.5281 | 0.8090 | 0.4408 | 0.8099 | 0.3718 | 0.7884 |

### A.8   Case Study of the Searched Cross-features.

Since only the ML1M dataset offers the specific meaning of each field of features, we further study the searching results from PROFIT on it. There are five fields in ML dataset, including `UserID`, `MovieID`, `UserGender`, `UserAge` and `UserOccupation`. During the progressive search procedure, our PROFIT learns the architecture parameter order by order. When learning the first-order features' architecture parameters, `UserGender` ranks lowest; as for the second-order feature, {`UserID`, `MovieID`} ranks first, which is in accord with the collaborative filtering methods that focus on the user-movie matching [32]. When learning third-order features' weights, {`UserGender`, `UserAge`, `Occupation`} ranks very high out of all third-order features.

Furthermore, we study the performance of reserving specific orders of cross-features and present the results of the ML1M dataset in Figure 4.2. We can observe that reserving higher-order cross-features and removing lower-order ones always worsen the prediction performance, which is consistent with the above-mentioned order-priority property.