# OpenReview forum: "Progressive Feature Interaction Search for Deep Sparse Network"
_NeurIPS.cc/2021/Conference — NeurIPS 2021 Poster_

### Official Review · Reviewer_B6yE · 2021-07-14

**Rating:** 7
**Confidence:** 3

**Summary:**

In this paper, the authors study the problem of deep sparse prediction based on NAS.
In order to reduce the search cost, they propose a distilled search space, which is a low-rank approximation of the full space.
They also propose a progressive differentiable search to capture the order-priority in sparse prediction.
Numerous experiments are conducted to demonstrate the effectiveness of the proposed method.


**Ethical Concerns:**

None.

**Limitations And Societal Impact:**

Please refer to the main review.

**Main Review:**

In general, the paper is written well and easy to follow. However, I still have several concerns that the authors need to address:
1. **Questions about distilling the search space.**
(1) As Figure 2 shows, the authors only use the second-order interaction to conduct the experiment, so does the higher-order interaction also have similar properties? Please give some theoretical proofs or display the experimental results;
(2) Equation 5 is very confusing. Please explain the following questions in detail:
    - Why A can be approximately equal to this form?
    - What conditions are needed to meet this approximation and how about the error?
    - What is the specific meaning of R, and how to set the value of R in the experiments.
    - Please give the detailed proof and deduction process of equation 5.

2. **Questions about efficiency.**
(1) In lines 213-214, the authors point that only N_0 important weights are retained. What is the basis for the selection of N0, what is the specific value, and how the selection of N_0 affects the accuracy and effectiveness? Please give more detailed experiments and explanations.
(2) In lines 262-263, the authors point that the experiment only evaluated the inference time and the size of the models. I hope that the author should also give the computation cost during training and inferences, such as FLOPs or MACs or other indicators to illustrate the efficiency.
(3) The author just compared only one NAS-based baseline in terms of efficiency. More challenging baselines are needed, such as the methods based on pruning [1,2].

3. **Questions about experiments.**
(1) The performance improvement in Table 1 seems limited. For example, the AUC in the Criteo dataset is only increased by 0.0001 compared with AFN.
(2) In lines 112-113, the author mentioned that some similar NAS-based methods usually require a large search cost, but the author did not compare the search efficiency with these methods in the experiment.

[1] Liu, Siyi, et al. "Learnable Embedding Sizes for Recommender Systems." arXiv preprint arXiv:2101.07577 (2021).
[2] Zhao, Xiangyu, et al. "Memory-efficient embedding for recommendations." arXiv preprint arXiv:2006.14827 (2020).


**Time Spent Reviewing:**

4

---

> ### Author Response · Authors · 2021-08-10
> **Response to Reviewer B6yE**
>
> Thanks for your valuable comments. We have split your comments into the following three questions and answer them one by one. We hope the following responses can address your concerns.
>
> **Q1. Questions about distilling the search space.**
>
> **Response:** As for your mentioned first point, *the authors only use the second-order interaction to conduct the experiment, so does the higher-order interaction also have similar properties*", we also conduct the same experiment on 3rd-order feature interactions. We have found very similar observations, and thus, we only show the 2nd-order. Thanks for your important advice, and we will add it in the final version.
>
> As for your mentioned second point, "*Equation 5 is very confusing*",  we would like to offer more detailed clarifications of Eqn (5). The unclear presentation is due to that we use the same symbol $\beta$ to refer to both the approximated tensor ($\beta^o$) and the decomposed tensor ($\beta^r$). Actually, the decomposed tensor is learned by gradient descent. Since our early empirical studies have shown there is a low-rank property, we propose to approximate $A$ by CP decomposition.  Although it is hard to provide theoretical proof and bar, we can provide a more intuitive explanation. The original space parametrized by $A$ has ignored that there is redundancy in modeling different cross-features. The CP decomposition capture the connections between important cross-features, such as that important low-order cross-features can be combined to build important high-order cross-features. Therefore it can help distill the search space.
>
>
> **Q2: Questions about efficiency.**
>
> **Response:**
>
> (1) In our experiments, to select $N_0$, we choose a simple yet effective manner by tuning a reserving ratio of every-order cross-features. We search the ratio from $0.1$ to $1.0$ with a step $0.1$.  This step will not cost too much time compared with the former step of searching architecture parameters.
>
> (2) You are right that we should report FLOPs or MACS, and therefore, we calculate the FLOPs on two utilized relatively larger datasets, as the following table shows.
>
> ```
> |-----------Criteo--|--Avazu--|
> | DeepFM |  22.7M   |  22.5M  |
> | AutoFIS|  22.8M   |  22.6M  |
> | PROFIT |  15.8M   |  18.1M  |
> ```
>
> (3)  AutoDim [1] and PEP [2] try to assign different sizes of embeddings to features largely according to the popularity of features. By assigning smaller embedding sizes to those tail features, the storage of the embedding table becomes smaller. It reduces the embedding parameter rather than handling cross-features. In fact, the results shown in AutoDim or PEP do not show efficiency improvement. They focus on whether the model can be stored or not in the edge devices. Therefore, these two works focus on a different aspect compared with our work. Our work can actually adopt AutoDim or PEP as the embedding layer and then feed feature embeddings to the feature-interaction layer.
>
>
>
> **Q3: Questions about experiments.**
>
> **Response:**
>
> (1) As widely acknowledged by existing works, including AutoInt, AFN, DeepFM, AutoInt, etc., achieving 0.001-level improvement on most metrics and achieve no-worse performance on a few metrics can be regarded as promising results. In our responses to Reviewer ewUq, we list a table that summarizes the recent works' improvement compared with the best baseline. Please refer to the table for checking.
>
> (2) We have presented the searching curves in Figure 6, comparing the search efficiency between our method and the existing method. We can observe that our method is significantly faster.

---

### Official Review · Reviewer_Uczd · 2021-07-15

**Rating:** 7
**Confidence:** 4

**Summary:**

This paper targets applying NAS to Deep Sparse Network (DSN) domain. To search for a good DSN, previous methods that apply DARTS based method to search an encoding vector, where each element represents the probability to select feature-interaction. Due to the interaction grows in a factorial way, the search parameters will soon become intractable. This work proposes to decompose this vector into multiple low-rank ones, to reduce the search space dramatically.

Experiments on various datasets show a clear improvement comparing to the previous baseline AutoFIS. I would recommend accepting this one but since I am not really familiar with the DSN domain, I might change my mind based on other reviews.

**Limitations And Societal Impact:**

Nothing special.

**Main Review:**

# Strength

+ Interesting idea to consider order-priotity property in DSN
+ Reasonable toy-case demonstration, showing that the matrix of second-order interaction exploits a low-rank property comparing to randomly generated matrices, this makes the proposed low-rank method conceptually meaningful.
+ This method reduce the search parameters from O(M^O) to O(MO).
+ Results shows constant improvement over the previous baseline AutoFIS.

# Weakness

- Motivation
This work observes the previous baselines (AutoFIS and another one) that applies NAS technique to deep sparse network have extreme high search cost. However, I read the AutoFIS paper that uses DARTS as search strategy, where they only need maximum 12 min in their Table 1. To me this is not extremely high cost. What's the value of your work in terms of this?

In addition, in Line 40 you go back to discuss the human-designed DSNs that has poor computational cost accuracy trade-off. As your method is not the first one apply NAS into DSN domain, to me, what's more interesting is to discuss the methods with NAS. Are they also suffering from this? Could you provide concrete numbers rather than `inferior efficiency`?

- Comparision with other methods
In the section 2.2, What's the difference in terms of methodology of other DSN with NAS baselines with your PROFIT? In other words, what makes your method search fast? If I understand correctly, it's your contribution to formulate the space encoding into low-rank PCA-like vectors to reduce the cost. Anyway, adding this info here might further enhance this section. And you probably want to emphasize this in the introduction as well.

- order-priority property, any justification?
The results shown in fig 2 only shows comparing to random matrices. This may not be sufficient to show the real order-priority property isn't it?

## minor issues
- AutoDSN search space
The paper claims C is relatively large that burdens the search cost, isn't this true for any NAS problem? DARTs for example has a space with billions of architectures. I remotely sense that this search space is somehow different with the traditional one, as the feature interactions may not be simplified by the weight sharing approach. But this is not really clear to me, hope the authors can clarify further why this space is a NP-hard problem and why the traditional NAS approach cannot resolve this. I saw section 3.1 answers part of my problem. Please correct me if I understand anything wrong.


- Writing issues
Page 2 Line 38 - 39
> However, these works require extremely large search cost....

This line is redundant with the previous sentence
> these methods are suffering from high search
38 costs, resulting in limited prediction performance and application value


**Time Spent Reviewing:**

6

---

> ### Author Response · Authors · 2021-08-10
> **Response to Reviewer Uczd**
>
> Thanks for your valuable comments. We have split your comments into the following seven questions and answer them one by one. We hope the following responses can address your concerns.
>
> **Q1. About the search cost of previous baselines.**
>
> **Response:** Thanks for this comment. We would like to have a clearer discussion about the efficiency of AutoFIS. First, the maximum search cost is 128 minutes, as reported in the original paper (https://dl.acm.org/doi/abs/10.1145/3394486.3403314). Second, more importantly, this search cost is calculated when the maximum order is set to three, which sets constraints to the utility. When increasing the order, the search cost will increase largely, which can also be observed from the increased search cost from 2nd-order to 3rd-order. Last, the authors of AutoFIS did not provide the details of their hardware platform, and thus it is hard to compare the numbers directly.
>
> As for your mentioned second point, "*In addition, in Line 40 you go back to discuss the human-designed DSNs that has poor computational cost accuracy trade-off. As your method is not the first one apply NAS into DSN domain, to me, what's more interesting is to discuss the methods with NAS. Are they also suffering from this?*", we would like to clarify that AutoFIS is the state-of-the-art method that applies NAS into the DSN domain. We provide a detailed comparison between it and our method in experiments.
>
> **Q2: Comparison with other methods In the section 2.2, What's the difference in terms of methodology of other DSN with NAS baselines with your PROFIT?  If I understand correctly, it's your contribution to formulate the space encoding into low-rank PCA-like vectors to reduce the cost. Anyway, adding this info here might further enhance this section. And you probably want to emphasize this in the introduction as well**
>
> **Response:** Thank you for this comment. In Section 2.2, we have mentioned three recent approaches that propose to adopt NAS to help explicitly search for powerful cross-features, AutoCross, AutoGroup, and AutoFIS, out of which AutoFIS is state-of-the-art and also the only open-source method. Compared with AutoFIS, we propose a more proper search space with a low-rank approximation that reduces the search cost largely and a progressive search strategy obtaining effective models. You are right that we should emphasize it in both Section 2.2 and the introduction, and we will add it in the final version.
>
> **Q3: Could you provide concrete numbers rather than inferior efficiency?**
>
> **Response:** Thanks for this comment. Inference time is hardware-dependent; therefore, we only say "inferior efficiency" in the introduction, and the exact time in our experimental platform can be found in Figure 3.
>
> **Q4: Justification on order-priority. The results shown in fig 2 only shows comparing to random matrices. This may not be sufficient to show the real order-priority property isn't it?**
>
> **Response:** Thanks for this comment. The random matrix can also be seen as the initialization of the interaction matrix. After bi-level optimization of interaction weights, the singular values become "parameter matrix" in Figure 2(a). Thus, we can see that singular values become skew during the iteration. We will add more details and clarify this point in the final version.
>
> **Q5: AutoDSN search space The paper claims C is relatively large that burdens the search cost, isn't this true for any NAS problem? DARTs for example has a space with billions of architectures. I remotely sense that this search space is somehow different with the traditional one, as the feature interactions may not be simplified by the weight sharing approach.**
>
> **Response:** Thanks for this valuable comment.
>
> - In DARTS, the C corresponds to the number of operations of neural networks, which is always on a scale of 3 to 7.
> - In our AutoDSN problem, the C refers to the number of features, which has a scale of 10 to 39 in benchmark datasets.
>
> Note that the space increases exponentially. Therefore, compared with the search space in DARTS paper, our space is therefore much larger than that of DARTS. Considering the real-world application where the searching cost and inference efficiency are widely concerned, this paper aims to address the efficiency-utility dilemma by proposing a more efficient and effective search space.
>
> **Q6: But this is not really clear to me, hope the authors can clarify further why this space is a NP-hard problem and why the traditional NAS approach cannot resolve this. I saw section 3.1 answers part of my problem. Please correct me if I understand anything wrong.**
>
> **Response:** Thanks for this comment. Fundamentally, enumeration is needed to solve the above search problem, which means it is NP-Hard. We need to assign a scalar for each possible choice on the exponentially-increasing space, which is doable for DARTS' space but not for ours (see our above response).
>
> **Q7: Writing issues.**
>
> **Response:** Thanks for pointing out these writing issues. We have conducted careful language checking and will fix writing issues in the final version.

---

> > ### Comment · Reviewer_Uczd · 2021-08-25
> > **Follow up questions**
> >
> > Thanks the authors for the response. I still have concerns regarding these questions:
> >
> > # Q1.
> >
> > Yes I can see the cost increase when the order increase. If you don't mind would you provide a comparison of your work and AutoFIS here or point me to the specific lines in your paper rather than a rough range? It's kind of difficult to find them line by line.
> >
> > # Q3
> >
> > Still in Figure 3 you compare the efficiency in a fair comparison. This contradicts to your earlier reply "Inference time is hardware-dependent; therefore, we only say "inferior efficiency" in the introduction". Also, from the figure 3, I saw your final model is only marginally better than xDeepFM in 4 out of 6 scenarios. It feels like the reason to not show concrete number because of this would undermines the usefulness of the PROFIT method.
> >
> > # Q4
> >
> > Sorry but I cannot see why the skew singular values can link to the significant order-priority. Would you provide more explanations?
> >
> > # Q6
> >
> > What's the enumeration you are referring to? Could the authors be more specific in their answers? I would not like to guess the meaning out of them.
> >
> > For example, in DARTS original space, billions of architectures are efficiently stored in their vector formulatization. Even if you have a much larger search space. If you have N nodes, the total representation is only N*C which is 4 * 7 in DARTS case, and N * 39 in your case. Why is this not doable?

---

> > > ### Author Response · Authors · 2021-08-31
> > > **Response to follow up questions of Reviewer Uczd**
> > >
> > > Dear Reviewer, thanks for your reply. We would like to provide the following responses to address your concerns.
> > >
> > > **Q1: Yes I can see the cost increase when the order increase. If you don't mind would you provide a comparison of your work and AutoFIS here or point me to the specific lines in your paper rather than a rough range? It's kind of difficult to find them line by line.**
> > >
> > > **Response**: AutoFIS uses a full space while our PROFIT utilizes a distilled search space (at line 170, we introduce how to distill the space; at line 300, we show the experimental results with different spaces). We have shown the search procedure in Figure 6 and the final results in Table 3 (line 300-308). We can observe that searching in the distilled space can efficiently reach a promising performance. However, when searching in the full space is far slower.
> > >
> > > **Q3: Still in Figure 3 you compare the efficiency in a fair comparison. This contradicts to your earlier reply "Inference time is hardware-dependent; therefore, we only say "inferior efficiency" in the introduction". Also, from the figure 3, I saw your final model is only marginally better than xDeepFM in 4 out of 6 scenarios. It feels like the reason to not show concrete number because of this would undermines the usefulness of the PROFIT method.**
> > >
> > > **Response:** Sorry for causing such a misunderstanding.
> > >
> > > 1)  First, in Figure 3, PROFIT achieves steady improvement against xDeepFM on three datasets and four metrics (AUC, LogLoss, Model Size, and Inference Time).
> > > 2)  Second, “Inference time is hardware-dependent”, as says, means the specific number of inference time varies on different hardware and different experimental setups. Thus, we have not provided the specific number in the introduction.
> > >
> > > We will provide the exact numbers in Figure 3 in the final version.
> > >
> > > **Q4: Sorry but I cannot see why the skew singular values can link to the significant order-priority. Would you provide more explanations?**
> > >
> > > **Response:**  We would like to clarify there is no relation between skew singular values and order-priority, i.e., there is no direct relationship between Figure (a) and (b).
> > >
> > > - Skew singular values in Figure 2 (a) shows there is redundancy in the architecture matrix for the second-order cross-feature (i.e., **intra order**).
> > > - In Figure 2 (b), we present the correlations between 2nd-order and 3rd-order cross-features (i.e., **between order**). We can observe that a good 2nd-order cross-feature is more likely to generate a good 3rd-order cross-feature.
> > >
> > > In Figure 4, we have also presented the performance of utilizing different orders' cross-features. These observations support the order-priority property: lower-order cross-features are likely to be more vital compared with higher-order ones, and higher-order features' quality can be relevant to their de-generated low-order ones.
> > >
> > > **Q6: What's the enumeration you are referring to? Could the authors be more specific in their answers? I would not like to guess the meaning out of them. For example, in DARTS original space, billions of architectures are efficiently stored in their vector formularization. Even if you have a much larger search space. If you have N nodes, the total representation is only N*C which is 4 * 7 in DARTS case, and N * 39 in your case. Why is this not doable?**
> > >
> > > **Response:** The enumeration means to enumerate all possible cross-feature sets in the feature-interaction layer, the full search space in DSN problem.
> > > In the problem of DSN, AutoFIS is the recently proposed DARTS-based solution, as we introduce in Section 3.1 (line 138-165). The complexity of using DARTS to handle $O$ orders and $M$ fields, by assigning a parameterized vector with a size of $C_M^1+C_M^2+...+C_M^O$, approximate to $M^O$. It explains why AutoFIS's searching process is slow when setting $O=3$: memory cost is approximately $39^3*E$ where $E$ is the feature's embedding size.
> > >
> > > In the original DARTS, differently, the search space is $N$ mixtures of $C$ candidate operations. The corresponding cost is approximately $N*C$.

---

> > > > ### Comment · Reviewer_Uczd · 2021-09-01
> > > > **Thanks for the clarification and I increase my score to accept**
> > > >
> > > > As titled.

---

### Official Review · Reviewer_ewUq · 2021-07-16

**Rating:** 6
**Confidence:** 4

**Summary:**

This paper proposes a method for searching for the interaction of the feature columns in a CTR prediction application. The paper title suggests an application for deep sparse networks but it doesn’t seem to be explicitly tied with the requirement of the input being sparse. To this end, the paper proposes an iterative algorithm that incrementally learns the interaction tensor from lower rank to higher rank, and from lower order to higher order. Empirically it observes some benefits compared to current competitive methods on Criteo, Avazu and ML1M datasets.

**Main Review:**

### Strengths
- The paper is clearly written.
- The proposed method is clear and relatively simple.
- It seems to outperform existing methods on several benchmarks.

---------------------------------------------
### Weaknesses
- **Significance of results:** While the proposed method almost achieves the best result across the board, I am not sure how significant it is. If we look at the AUC score, then it only improves upon the second best by 0.0002, 0.0027, 0.0008. I am no expert on these benchmarks but it still seems too small of a difference. How much difference does this bring in practice?
- **Limited scope and comparison to a more general architecture like Transformer:** As I mentioned in the summary, the paper focuses on the application of CTR prediction using a deep sparse network, but there is nothing in the method that actually relies on the input feature being sparse. In other words, the proposed method can be applied to anything that aims to look for high-order interactions among different dense features, but the scope is intentionally narrowed down for no reason. The compared methods are all densely populated in a very narrow domain. The author could also compare it with more general architecture like Transformers. One layer of the transformer allows pairwise interactions, and more layers can allow higher-order interactions. It is unintuitive what would be the benefit to directly learn an interaction tensor via the proposed approximation, as opposed to a more general method that involves attention.
- **Entangling rank vs. order:** I found it strange that the paper entangles the two concepts of rank and order. According to Algorithm 1, it is unclear why we want to increase both the rank and the order by 1. Alternatively, you could increase the order by 1 but have a rank-3 approximation throughout. But the current algorithm learns one vector at a time, so it means that the rank is also the same as the order.
- **Comparison to asymmetric low-rank space:** There can be an additional comparison in between “full space” and “PROFIT’s space,” where the space is low rank but asymmetric. In other words, it is unclear what benefit the assumption of symmetric tensors brings.

---------------------------------------------
### Conclusion
In conclusion, the contribution of this paper is clear. But I am not sure about its significance in terms of the results as well as the scope of the paper. Therefore, my initial rating is slightly below the bar.

Update: the rebuttal addresses most of my concerns and I increased the score from 5 to 6.

**Time Spent Reviewing:**

4

---

> ### Author Response · Authors · 2021-08-10
> **Response to Reviewer ewUq**
>
> Thanks for your valuable comments. We have split your comments into the following five questions and answer them one by one. We hope the following responses can address your concerns.
>
> **Q1. While the proposed method almost achieves the best result across the board, I am not sure how significant it is. If we look at the AUC score, then it only improves upon the second-best by 0.0002, 0.0027, 0.0008.**
>
> **Response:** Thanks for pointing it out. In the evaluation of these benchmark datasets, it is widely acknowledged by existing works, including AutoFIS, AutoInt, DeepFM, etc., that a 0.001-level improvement on Logloss/AUC can be claimed as significant. Here we list a table that summarizes the recent works' improvement compared with the best baseline as follows. We can observe that on these utilized benchmark datasets, the improvement achieved by our proposed method makes sense.
>
> ```
> |--------------- Criteo --- | ----- Avazu ---- | ----- ML1M  ----|
> |----------|  AUC  |  Loss  |   AUC   |  Loss  |   AUC  |  Loss  |
> | AutoFIS  |0.0019 | 0.0026 |  0.0016 | 0.0016 |   N/A  |   N/A  |
> | AutoInt  |0.0052 | 0.0053 | -0.0060 | 0.0005 | 0.0008 | 0.0017 |
> | xDeepFM  |0.0014 | 0.0049 |   N/A   |   N/A  |   N/A  |   N/A  |
> | AFN      |0.0004 |-0.0008 |  0.0005 | 0.0003 | 0.0006 | 0.0015 |
> | DeepFM   |0.0027 | 0.0030 |   N/A   |   N/A  |   N/A  |   N/A  |
> ```
>
> We also calculate the STD values in our PROFIT of different running instances with various random seeds. We then find that almost all of them are no larger than $0.0002$ among three datasets and two metrics.
>
> **Q2: How much difference does this bring in practice?**
>
> **Response:** In real-world applications, both the accuracy and the inference efficiency should be considered. Our method not only achieves promising accuracy but also reduces the inference time. Therefore, it has high application value in practice.
>
>
> **Q3: Limited scope and comparison to a more general architecture like Transformer.**
>
> **Response:** Thanks for pointing it out. For your mentioned first point, "*there is nothing in the method that actually relies on the input feature being sparse. In other words, the proposed method can be applied to anything that aims to look for high-order interactions among different dense features, but the scope is intentionally narrowed down for no reason*", we would like to provide a clearer explanation of why we search for the high-order feature interaction.
>
> - Actually, the deep sparse network architecture, with a lot of human-crafted designs including AFN, AutoInt, xDeepFM, etc., serves as a standard solution for CTR prediction, which is an important research field.
> - Our work focuses on addressing the limitation of these models via neural architecture search. Feature interaction is the most mainstream and widely-used operation for handling sparse features, while for dense features, there exist other suitable operations such as numerical operations on the distribution.
>
> As a result, the position of the submission is motivated by the expert knowledge of this area which may not be proper for other fields.
>
> For the second point, "*The author could also compare it with more general architecture like Transformers*", we would like to provide a clearer discussion on the compared AutoInt model. In AutoInt, the feature-interaction layer is designed to stacked multi-layer self-attention architectures. As mentioned by the authors of AutoInt, they propose to stack the adapted Transformer architecture in feature-interaction layers for handling and modeling higher-order cross-features. Therefore, AutoInt serves as a general method that involves attention.
>
>
> **Q4: Entangling rank v.s. order.**
>
> **Response:** We are sorry for our unclear presentation. This is due to that in Eqn (5), we use the same symbol $\beta$ to refer to both the approximated tensor ($\beta^o$) and the decomposed tensor ($\beta^r$). In Algorithm 1, it refers to the approximated tensor. Therefore there is no conflict between rank and order.
>
>
> **Q5: Comparison to asymmetric low-rank space.**
>
> **Response:** Thanks for this comment. Here the symmetric CP decomposition is naturally motivated by the fact that feature interaction is perturbation invariant, i.e., $f_1 \odot f_2 = f_2 \odot f_1$.
>
> We follow your advice and conduct experiments with an asymmetric low-rank space. The results show a 0.001-level decrease of accuracy on average compared with the symmetric low-rank space. This demonstrates the rationality of the symmetric low-rank space, and we will add it in the final version.

---

### Official Review · Reviewer_5u8P · 2021-07-20

**Rating:** 7
**Confidence:** 3

**Summary:**

This paper proposes a neural architecture search approach, PROFIT,  for deep sparse networks.  PROFIT introduces a distilled low-rank search space and a progressive search algorithm from the lower orders. The proposed method is evaluated on three benchmark datasets and the authors conduct extensive ablation study and discussion on the method. The paper is well written and very easy to follow even for people without a NAS background.



**Limitations And Societal Impact:**

Yes.

**Main Review:**

1. Originality: Are the tasks or methods new? Is the work a novel combination of well-known techniques? (This can be valuable!) Is it clear how this work differs from previous contributions? Is related work adequately cited?

Neural architecture search is no longer a new task. However, this work proposes a very reasonable solution to address the high cost of neural architecture search. It also puts the focus on interpretable sparse architecture search, which seems to be a novel and interesting perspective.  The discussion with the previous work is extensive and also easy to follow.

2. Quality: Is the submission technically sound? Are claims well supported (e.g., by theoretical analysis or experimental results)? Are the methods used appropriate? Is this a complete piece of work or work in progress? Are the authors careful and honest about evaluating both the strengths and weaknesses of their work?

The submission is technically sound.  The authors provide an extensive evaluation of three benchmark datasets. For example,  they provide motivating plots (e.g., Figure 2) to help the readers understand their approach, which is very helpful.  The method itself is kind of straightforward, but it's well-motivated and also quite effective.

3. Clarity: Is the submission clearly written? Is it well organized? (If not, please make constructive suggestions for improving its clarity.) Does it adequately inform the reader? (Note that a superbly written paper provides enough information for an expert reader to reproduce its results.)

This paper is very well written and very easy to follow. Even readers outside the NAS community can understand it.  The paper structure is clear.  There are a few typos in the paper, though. For example, the Aycronm PROFIT directly pops up in the abstract without reference. Line 260 extra space in the word "evaluate".

4. Significance: Are the results important? Are others (researchers or practitioners) likely to use the ideas or build on them? Does the submission address a difficult task in a better way than previous work? Does it advance the state of the art in a demonstrable way? Does it p
rovide unique data, unique conclusions about existing data, or a unique theoretical or experimental approach?


The results seem important and  it also achieves the state-of-the-art performance on three datasets.    It might be inspiring for others in the NAS field  and the deep  sparse networks.

**Time Spent Reviewing:**

5

---

> ### Author Response · Authors · 2021-08-10
> **Response to Reviewer 5u8P**
>
> Thanks for your positive comments. As for your mentioned typos, we will conduct careful language checking in our final version.

---

### Official Review · Reviewer_hemK · 2021-09-16

**Rating:** 6
**Confidence:** 4

**Summary:**

The authors consider the problem of designing accurate and computationally efficient deep sparse networks (DSNs), which are an important problem in applications with sparse features such as click-through rate or movie recommendation. The authors propose a new approach based on neural architecture search (NAS) using two techniques: distillation of the search space (of cross-features), and a progressive search algorithm. They show that their approach leads to improvements in both accuracy and inference time, on three datasets.

**Ethical Concerns:**

There are no ethical issues.

**Limitations And Societal Impact:**

The authors did not explicitly discuss limitations, as far as I could tell. Societal impact was addressed in the checklist.

**Main Review:**

## Strengths
- Machine learning problems with sparse features have popular applications, and so empirical improvements could have high impact.
- The two parts of the algorithm, distillation of the search space and progressive search over feature orders, appears to be well-suited to the problem.
- The results are strong. The authors pick a varied list of first-, second-, and high-order methods for comparison, as well as AutoFIS, the only other NAS-based DSN work that is open-source.  Table 1 and Figure 3 seem convincing.

## Points that could be improved
- Novelty/Impact. NAS has been applied to DSNs at least twice before, as cited by the authors. The ideas of distillation, and progressive search, have been known in the NAS community, although not used the same way as this work (e.g. [1] uses a technique to compress a DARTS-like search space). The main novelty/impact is combining and applying these ideas to the DSN setting, and showing that it leads to strong empirical performance. The technique is tailored to this application.
- Reproducibility: the authors give specific implementation details in the appendix, which is good. However, they do not release their code? They mention in the checklist (twice) that the code is released, but I could not find it. Can somebody correct me if I am wrong? Reproducibility is an important consideration, both in the field of NAS where reproducibility is a problem, and also for papers in which the largest contributions are empirical and application-specific, such as this paper. For example, open-source code can lead to higher impact.
- The authors conduct ablation studies for both the progressive search strategy and the distillation strategy, as well as a few more in the appendix (different ranks, output layer, embedding size), which is excellent. However, it would be helpful to put these ablation studies onto one plot, to better answer the question: which of the main techniques helped more (e.g. distillation or progressive search)?
- The paper is written okay. There are some places that could be made clearer. Section 3.3. seems a bit out of place, and related work shows up in different places, and for example, the footnotes on page 7 could me moved to the appendix.

## Overall
The paper is okay. The authors study DSNs using NAS, and the techniques are not extremely novel, but the execution of the techniques is well-done and leads to strong performance. Reproducibility could be improved. The decision could go either way, and currently I lean towards weak reject.

[1] https://arxiv.org/abs/1906.02869


**Time Spent Reviewing:**

4

---

> ### Author Response · Authors · 2021-09-18
> **Responses to the newly-added comment of Reviewer hemK**
>
>  We sincerely thank the newly-added comment for helping further improve our paper. To address the concerns, we make the responses one by one as follows.
>
> **Q1:Novelty/Impact. NAS has been applied to DSNs at least twice before, as cited by the authors. The ideas of distillation, and progressive search, have been known in the NAS community, although not used the same way as this work (e.g. [1] uses a technique to compress a DARTS-like search space). The main novelty/impact is combining and applying these ideas to the DSN setting, and showing that it leads to strong empirical performance. The technique is tailored to this application.**
>
> **Response:** We would like to clarify the difference between our method and [1]. Actually, [1] is based on compressed sensing, while our method is based on low-rank approximation. Therefore, although these two works both conduct the space distillation, the fundamental assumptions (vector space v.s. matrix space) are different. Please also have a check over [2, 3].
> In addition, in our earlier exploration of this work, we have some preliminary results that verify that [1] is worse than our method in the DSN problem. We will also include this baseline in the final version.
>
> [2] Exact matrix completion via convex optimization. FCM. 2008
>
> [3] Compressed sensing, TIT 2006
>
> **Q2:Reproducibility: the authors give specific implementation details in the appendix, which is good. However, they do not release their code? They mention in the checklist (twice) that the code is released, but I could not find it. Can somebody correct me if I am wrong? Reproducibility is an important consideration, both in the field of NAS where reproducibility is a problem, and also for papers in which the largest contributions are empirical and application-specific, such as this paper. For example, open-source code can lead to higher impact.**
>
> **Response:** Thanks for kindly reminding us. We now release our codes in this anonymous link: https://github.com/NeurIPS-AutoDSN/NeurIPS-AutoDSN.
>
>
> **Q3:The authors conduct ablation studies for both the progressive search strategy and the distillation strategy, as well as a few more in the appendix (different ranks, output layer, embedding size), which is excellent. However, it would be helpful to put these ablation studies onto one plot, to better answer the question: which of the main techniques helped more (e.g. distillation or progressive search)?**
>
> **Response:** Thanks for this valuable comment. You are right that putting results together will lead to some insightful comparison among different ablation studies. As shown by the results, the distillation is far more important since the full space will result in a large search cost, which significantly worsens the final performance. We will adjust the figures to merge plots in the final version.
>
>
> **Q4:The paper is written okay. There are some places that could be made clearer. Section 3.3. seems a bit out of place, and related work shows up in different places, and for example, the footnotes on page 7 could me moved to the appendix.**
>
> **Response:** Thanks for this comment. We will re-organize the texts along with figures/tables to achieve a better presentation in the final version.

---

> > ### Comment · Reviewer_hemK · 2021-09-19
> > **Thanks for the response**
> >
> > I appreciate the authors addressing my comments.
> >
> > Q1: yes, I agree that the method of distillation from [1] and your work is different (although [1] was the first to apply distillation in NAS). The authors mentioned adding [1] as a baseline, and I agree that would be interesting.
> >
> > Q2: thank you for providing the code. This makes your work reproducible and makes me feel better about the paper overall.
> >
> > I also have a couple comments based on looking at the other discussions.
> >
> > - The experiments could be stronger if the authors compared their method to 3rd order AutoFIS, and the runtime of that is about 2 GPU-hours on Criteo. Although, I understand that the type of GPU was not specified in the AutoFIS paper, and the search space would be size 39^3 * (embedding size), which might make it harder to run.
> >
> > - I think the paper does not have a formal proof of the claims on lines 52 and 152 that “searching in this space is an NP-hard problem”. In the discussion phase, the authors only explained why running DARTS would take exponential time. It might be better to adjust those claims, or else to give a more formal argument for NP-hardness.

---

### Decision · Program_Chairs · 2021-09-27

**Decision:**

Accept (Poster)

**Comment:**

This paper proposes using NAS for learning the feature interaction space for deep sparse networks that are used for high dimensional sparse networks commonly encountered in recommendation systems. The discovered architectures have higher efficiency compared to the baselines.

During discussion period some points of confusion consistently came up regarding order-vs-rank and the fact that it was not explicitly clear that the search space in feature-interactions is several orders of magnitude bigger than that used in DARTS for CNNs due to the cardinality of the operators (7 or so in DARTs vs. 39 here).  This is a point that the authors can make clearer in the paper in the beginning itself. Otherwise the difficulty of the problem does not come about till later.

Reviewers have also pointed out various ways to make presentation better like not using forward references to acronyms before definition. Since the main contribution of this paper is the search space and the progressive search over feature orders and not the NAS search technique the authors should emphasize that part explicitly and play to the strengths of the paper in presentation.